# Reinforcement Learning and Heuristics for Hardware-Efficient Constrained Code Design

## Abstract

Constrained codes enhance reliability in high-speed communication systems and optimize bit efficiency when working with non-binary data representations (e.g., three-level ternary symbols). A key challenge in their design is minimizing the hardware complexity of the translation logic that encodes and decodes data. We introduce a reinforcement learning (RL)-based framework, augmented by a custom L1 similarity-based heuristic, to design hardware-efficient translation logic, navigating the vast solution space of codeword assignments. By modeling the task as a bipartite graph matching problem and using logic synthesis tools to evaluate hardware complexity, our RL approach outperforms human-derived solutions and generalizes to various code types. Finally, we analyze the learned policies to extract insights into high-performing strategies.

## 1 Introduction

**Reinforcement learning (RL)** has been successfully applied to numerous tasks in chip design. Works such as DRiLLS (Hosny et al., 2020) and Retrieval-Guided RL (Goliaei et al., 2024), have demonstrated RL's ability to optimize tasks like logic synthesis, which involves converting high-level hardware descriptions into optimized gate-level representations to minimize circuit complexity and improve performance. Similarly, Mirhoseini et al. (2021) introduced a graph-based RL methodology to optimize circuit placement, significantly reducing layout generation time compared to traditional methods.

Building on recent advances, we address a distinct challenge: **constrained code design**. Constrained codes restrict sequences or patterns in communication and data storage, ensuring they meet specific rules (e.g., avoiding certain bit patterns). Unlike typical digital design tasks where the logical specifications are fixed (e.g., an adder), our approach applies RL to determine valid and efficient codeword assignments that adhere to these constraints. Instead of simply optimizing existing logic, RL is used to dynamically generate the assignments in a lookup table structure, which presents a unique challenge for RL agents.

Constrained codes are crucial for ensuring data reliability and efficiency in many systems. Run-Length Limited (RLL) codes (e.g 8b10b) are implemented in standards like PCIe, USB, and Ethernet to prevent signal degradation by limiting long runs of similar bits (PCI-SIG, 2019). Similarly, Data-Bus Inversion (DBI) is used in memory interfaces like HBM and DDR to reduce power consumption (Hollis, 2009). Beyond these established uses, we propose applying constrained codes to further **compress ultra-quantized AI models**. Recent research on low-precision LLMs (Ma & all, 2024) shows promise for 1.58bit (ternary) precision models. Constrained codes could enhance this by combining multiple weights (e.g., ternary symbols) into more efficient encodings.

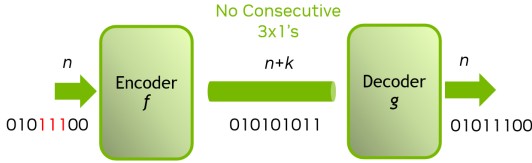

Figure 1: RLL Constrained Code Illustration

Formally, a constrained code maps an $n$-bit sequence to a longer $(n + k)$-bit sequence that adheres to specified constraints. The encoder, $f : 0, 1^n \rightarrow 0, 1^{n+k}$, maps the $n$-bit input to an encoded

output, while the decoder, $g : 0, 1^{n+k} \rightarrow 0, 1^n$, performs the inverse mapping, ensuring that $g(f(x)) = x$ for any input $x$. Figure 1 illustrates the encoder-decoder structure for an RLL code.

These functions are typically implemented as digital circuits, where logic complexity affects latency, power, and cost. From our encoder definition, there are $2^n$ input sequences to map to valid codewords. Let $v$ be the number of valid codewords where $v \geq 2^n$, and we must select a subset of size $2^n$ from $v$. The number of possible mappings is $\binom{v}{2^n}(2^n)!$. A straightforward approach is to define $f$ and $g$ by choosing from these assignments and implementing them using a lookup table (LUT). **Synthesis tools, however, cannot optimize these input mappings**, leaving this critical step to manual methods, which are often inefficient and time-consuming. With a **selected mapping**, synthesis tools can then convert these mappings into hardware, providing critical metrics such as gate count, area, and delay.

We demonstrate this process using Maximum Transition Avoidance (MTA) coding from GDDR6x memory, which maps binary data to PAM-4 symbols -3,-1,+1,+3 and avoids maximum transitions (-3 to +3) (Sudhakaran et al., 2021). The MTA code uses $n = 7, k = 1$ and has $v = 139$ valid codewords, making exhaustive synthesis of all $\binom{v}{2^n} 2^n!$ mappings impractical. Fig. 2 shows the gate counts from 1M randomized mappings versus a manually designed solution.

It becomes evident that there is a **significant gap** between purely **random assignments** and a **hand-crafted solution**. While hand-crafted designs can achieve better results, this process is both time-consuming and unscalable for larger problem instances without a clear algorithm. The vast solution space and inefficiency of random search highlight the need for automated optimization of codeword assignments.

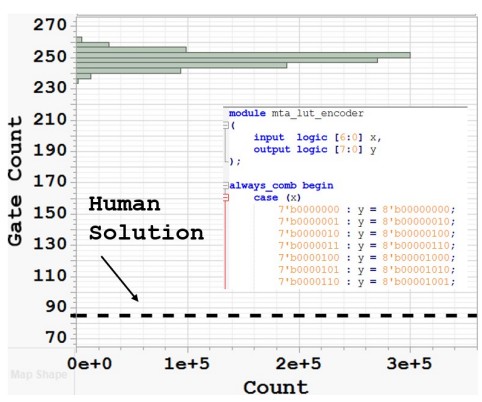

Figure 2: Randomized 1M LUT Mappings Gate Count Histogram for MTA 7-8 Code with LUT verilog code snippet

We propose a RL framework to automate exploration of the codeword assignment space. By framing the problem as a **combinatorial optimization task**, the RL agent iteratively learns optimal mappings using feedback on metrics like gate count, area, and latency. To improve efficiency, we incorporate a custom L1 similarity-based heuristic to prioritize promising mappings early. An external simulator evaluates the quality of each assignment, helping the agent navigate the large search space more effectively.

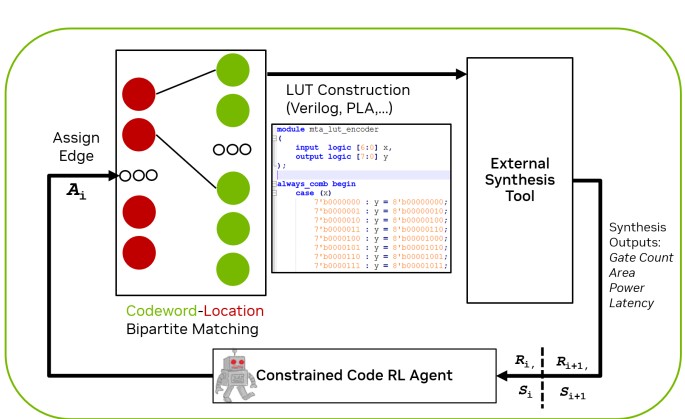

Figure 3: High-level RL Framework for our Bipartite Matching Codeword Assignment Problem

Figure 3 shows the framework flow, where the RL agent selects actions (mapping inputs to codewords), and the environment returns rewards based on synthesis metrics. This approach outperforms random search and classical optimization techniques.

## 2 BACKGROUND & RELATED WORK

Our codeword selection and assignment problem can be formulated as a *bipartite graph matching problem*, where the goal is to assign input sequences to codewords while satisfying specific constraints. Formally, a bipartite graph $G = (X, Y, E)$ consists of two disjoint sets of nodes: $X$, representing the *unrestricted domain* of input sequences (locations), and $Y$, representing the *restricted domain* of encoded outputs (codewords). Each node in $X$ corresponds to a potential input bit sequence, while each node in $Y$ corresponds to a valid encoded sequence. The edges $E \subseteq X \times Y$ represent valid assignments between input sequences and codewords, defining the possible mappings in the problem space.

To help illustrate this, consider a PAM-3 (Pulse-Amplitude-Modulation) encoder that convert binary data to ternary symbols (-1,0,+1). PAM3 encoders require $\nu = 3^s > 2^n$ where $s$ represents the number of PAM-3 symbols consisting of 2 bits each hence $s = \frac{n+k}{2}$. A simple encoder has $n = 3, k = 1, s = 2$. There are $\nu = 9$ code words

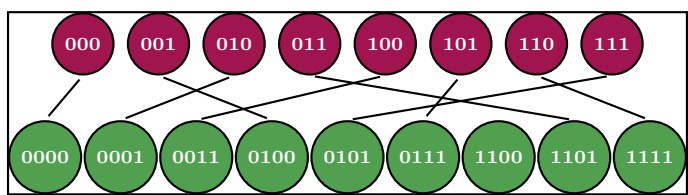

Figure 4: Bipartite Graph Representation of Location to Code Assignments for Simple 3b2s PAM-3 Code

which yields 1 unassigned code since $2^n = 8$. Figure 4 shows a bipartite graph formulation for the PAM 3b2s code. Note that the green colored nodes ($Y$) do not contain the "10" symbol.

This bipartite graph fully specifies our encoder and decoder functions $f, g$. Although this particular problem is simple enough to explore the full solution space consisting of $\binom{9}{8} \cdot 8! = 362,880$ unique LUT constructions, as $n$ and $\frac{n}{n+k}$ scale, the solution space explodes, giving rise to a wide range of logic complexity results.

Navigating large discrete solution spaces is a well-known challenge in combinatorial optimization. In the remainder of this section, we provide an overview of traditional and emerging approaches, highlighting their applications and limitations in solving our codeword selection and assignment problem.

### 2.1 CLASSICAL COMBINATORIAL OPTIMIZATION ALGORITHMS

For combinatorial optimization problems, Simulated Annealing (SA) (Kirkpatrick et al., 1983) and Monte Carlo Tree Search (MCTS) (Browne et al., 2012) are proven techniques. SA employs probabilistic exploration of solution spaces by allowing occasional moves away from gradients to escape local minima. While effective, SA can be slow to converge. In contrast, MCTS incrementally builds a search tree through random sampling and refines its exploration based on promising branches. Although successful in various combinatorial optimization and game-playing tasks, MCTS was deemed impractical for our problem due to the large branching factor and computational overhead.

### 2.2 LINEAR PROGRAMMING & SAT/SMT

Linear optimization techniques, including Linear Programming (LP) (Kuhn, 1955; Jonker & Volgenant, 1987), Integer Linear Programming (ILP) (Cunningham, 1976), and network flow algorithms (Edmonds & Karp, 1972), have been successfully applied to bipartite assignment and matching problems. These approaches rely on linear formulations and relaxations of the problem, seeking to optimize assignments between two sets (e.g., tasks and workers, or locations and resources) while minimizing or maximizing a given cost function. In the standard *assignment problem*, given two sets $X$ and $Y$ and a cost matrix $C$, where $C_{ij}$ represents the cost of assigning element $i \in X$ to element $j \in Y$, the objective is to minimize the total cost of assignment: $\min \sum_{i \in X} \sum_{j \in Y} C_{ij} x_{ij}$, subject to the constraints: $\sum_{j \in Y} x_{ij} = 1$ for all $i \in X$, and $\sum_{i \in X} x_{ij} = 1$ for all $j \in Y$, with $x_{ij} \in \{0, 1\}$. The *LP relaxation* relaxes this constraint to $x_{ij} \in [0, 1]$, making the problem tractable. These approaches assume that the assignment costs are readily available or precomputed,

which is infeasible given our need to invoke the synthesis tool for cost evaluations. In contrast, our RL framework samples potential assignments incrementally, allowing it to learn which assignments lead to better outcomes while minimizing the number of synthesis evaluations required.

In addition to linear optimization, the problem of codeword selection and logic complexity minimization has also been approached using SAT/SMT solvers. Recent work (Anonymous, 2024) has demonstrated their potential for small- to medium-sized codes. Rather than using a traditional LUT or matching approach, this work focused on generating an efficient circuit structure (e.g., Sum of Products) while asserting constraints on the total number of minterms (i.e., gates). While the authors do show their methods attain solutions competitive with human solutions for the MTA code, they faced challenges scaling to larger codes.

### 2.3 Neural Methods for Combinatorial Optimization, Code Design, and Logic Synthesis

Over the past several years, there has been considerable progress in Neural Combinatorial Optimization (NCO) using RL and Graph Neural Networks (GNNs). For example, Georgiev & Liò (2021) apply RL and GNNs to solve bipartite matching through a flow-based formulation, while Dwivedi et al. (2021) extend these approaches to problems like the Traveling Salesman Problem (TSP) and bipartite matching. Manchanda et al. (2023) further enhance NCO techniques by integrating meta-learning with RL to improve generalization across problem distributions. However, all these approaches compute assignment costs inline, contrasting with our method, which relies on external evaluations for cost calculations.

Related to the design of codes, Liao et al. (2020) and Miloslavskaya et al. (2024) use RL to construct polar codes for wireless communication. Their goal is to improve error-correction performance by framing bit selection as a sequential decision-making task. In contrast, our objective focuses on minimizing hardware logic complexity, which introduces different challenges. Furthermore, their method computes costs inline. Finally, Qin et al. (2023) use RL to develop fountain codes.

As noted in the Introduction, recent work such as DRiLLS and related frameworks have successfully applied RL to synthesis optimization tasks. These approaches focus on optimizing fixed logical structures like adders and multipliers by adjusting synthesis parameters. In contrast, our work tackles the problem of optimizing constrained code design, where the RL agent determines the logical structure itself, offering a distinct set of challenges and opportunities for optimization.

## 3 Methods & RL Framework

As outlined in Fig. 3, our RL framework models the codeword assignment problem as a Markov Decision Process (MDP). The state comprises of the current codeword assignments and synthesis tool metrics, and the action space includes potential codeword selections. The reward function evaluates hardware complexity, aiming to minimize metrics such as gate count, area, and power, guided by feedback from logic synthesis tools.

We formally define our objective function in Eqn. 1 where $A$ represents the bipartite graph assignments, $X$ and $Y$ are the nodes in the unrestricted and restricted domains respectively, and $\alpha, \beta, \gamma$ are all weighting terms based on common QoR metrics incuding area, delay, and power. The notation $X \times Y$ represents all the possible assignments in the Cartesian product.

$$\max_{A \subseteq X \times Y} \mathbf{QoR}(A) = \min_{A \subseteq X \times Y} \alpha \, \mathbf{Area}(A) + \beta \, \mathbf{Latency}(A) + \gamma \, \mathbf{Power}(A). \tag{1}$$

### 3.1 Reinforcement Learning Algorithms

To solve the bipartite matching problem using RL, we considered the type of learning approach (online vs. offline), the environment details (model-based vs. model-free), and the choice between value-based and policy-based networks. Additionally, our environment is **purely deterministic**. Given the same inputs, constraints, and parameters, the synthesis tools will consistently produce the same output, meaning each state-action pair will always lead to the same next state. Below, we outline the rationale behind our choices:

1. **Online** vs Offline Learning: We opted for **online learning**, where the agent interacts with the environment in real-time, dynamically adapting and refining its policy based on live feedback. This approach is particularly suitable for our problem given the fast solve times of the synthesis tools.

2. Model-Based vs **Model-Free**: Despite the deterministic environment, we chose **model-free** methods, which learn directly from interactions rather than relying on a pre-built model. Logic optimization has a highly non-linear solution space, which makes creating accurate performance models difficult. Given our fast solve times, we prioritized model-free methods for their simplicity and direct application. Model-free methods like Double-DQN, PPO, and **distributional RL** variants such as C25 and C11 (Bellemare et al., 2017) allowed us to focus on optimizing the policy without the additional complexity of maintaining an accurate model.

3. **Value-Based vs Policy-Based Network:** We explored both value-based methods, such as Double DQN, and C51, as well as policy-based methods, including A2C and PPO. We found the value-based methods to perform quite well with hyperparameter tuning with regards to both attained solutions and stability. Though we expected PPO to perform quite well due to its balance between exploitation and exploration, its performance was always worse than Double DQN.

As detailed in Section 4, most algorithms achieved similar final solutions in terms of value, but they differed in terms of convergence rates, stability, and other performance characteristics.

### 3.2 L1 Similarity & Heuristics

During our exploration of RL-based methods, we found that incorporating domain-specific heuristics was crucial in guiding the agent toward optimal solutions. Specifically, we used an L1 similarity metric on the location and codeword binary data to assess the benefit of assigning codewords to locations (see Algorithm 1). This heuristic proved to be a strong predictor of optimal assignments, improving performance and convergence. We also applied it with non-RL approaches such as greedy search and simulated annealing.

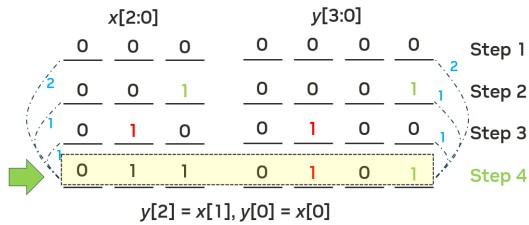

Figure 5: L1 Similarity Intuition

To illustrate the intuition behind our method, we present a simple example from the PAM3 3b2s code. We show the first four locations (000, 001, 010, 011) and the assignments to valid 4-bit PAM-3 codewords while avoiding restricted symbols (10). At the fourth step, we compute the L1 distances from the current location to previous locations and do the same for all available codewords to previously assigned codes. By choosing 0101, we reduce the logic for assignments: $y[2] = x[1]$ and $y[0] = x[0]$, resulting in no gates being required (just pass through).

In subsequent sections, we will compare the performance of RL-based methods against non-RL heuristics like simulated annealing and greedy search, both enhanced with L1 similarity, showing how RL improves solution quality and convergence rates.

### 3.3 Network Architecture

Figure 6 shows our network architecture which leverages a 3 layer Multi-Layer Perceptrons (MLPs). The MLP transforms the binary location and code nodes into embeddings with 256 dimensions, where the final layer outputs a 256-dimensional vector. For more details on the MLP architecture, please see the Appendix. While the MLP alone achieves competitive results, we experimented with a Graph Attention Network (GAT). The GAT operates on location and code nodes, learning attention coefficients. We found that the GAT accelerates convergence, but remains optional as L1 similarity combined with the MLP still performs well over longer training episodes as we will show in Sec. 4.

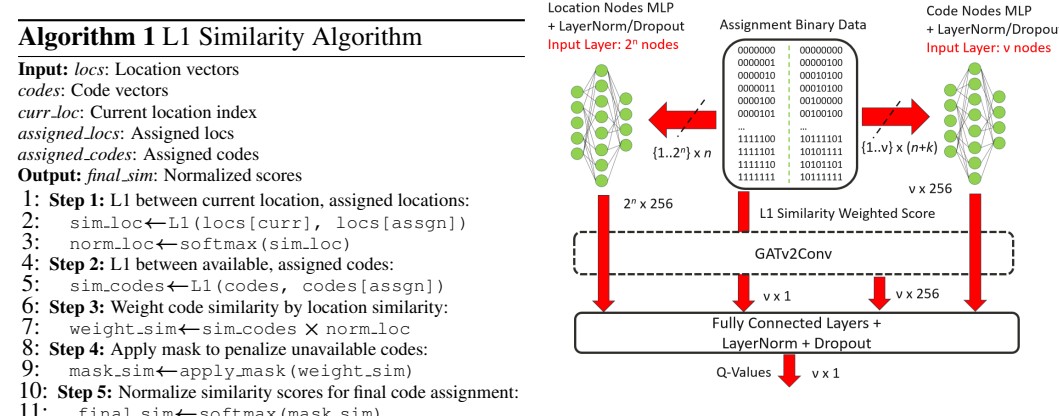

**Algorithm 1** L1 Similarity Algorithm

**Input:** *locs*: Location vectors
*codes*: Code vectors
*curr_loc*: Current location index
*assigned_locs*: Assigned locs
*assigned_codes*: Assigned codes
**Output:** *final_sim*: Normalized scores
1: **Step 1:** L1 between current location, assigned locations:
2:    `sim_loc←L1(locs[curr], locs[assgn])`
3:    `norm_loc←softmax(sim_loc)`
4: **Step 2:** L1 between available, assigned codes:
5:    `sim_codes←L1(codes, codes[assgn])`
6: **Step 3:** Weight code similarity by location similarity:
7:    `weight_sim←sim_codes × norm_loc`
8: **Step 4:** Apply mask to penalize unavailable codes:
9:    `mask_sim←apply_mask(weight_sim)`
10: **Step 5:** Normalize similarity scores for final code assignment:
11:    `final_sim←softmax(mask_sim)`

Figure 6: Network Architecture: MLP with Fully Connected Layers. Graph-Attention Layer shown as Optional

The final layers are fully connected layers which integrate the embeddings with the L1 similarity calculation. The resulting Q-values guide the agent's assignment decisions, leveraging both learned attention from the GAT and historical embedding similarities. We prove the benefit of each portion of the architecture through ablation studies in Section 4.

### 3.4 EPISODE PROGRESSION

An episode involves sequentially assigning code nodes to each location node from location 0 to $2^n - 1$. In our experiments with various ordering schemes, order had little effect on the output so we used the simplest scheme. At each step $i$, the external synthesis tool is called, incorporating the $i$ assignments made so far along with any "don't care" conditions. The synthesis tool then provides feedback, which we integrate into the reward function and state information. Fig. 7 illustrates the graph state at various stages of the episode. Initially, the graph contains mostly virtual edges, indicating possible assignments. As the episode progresses, these virtual edges are replaced by actual assignments, ending in a state where all location nodes are mapped to code nodes.

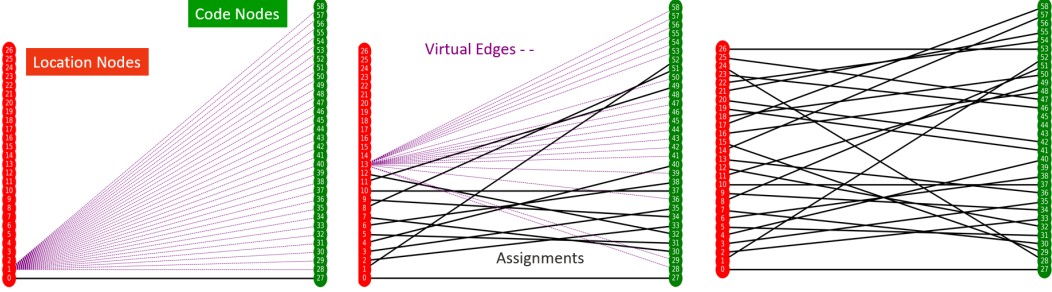

Figure 7: Graph Progression During Episode for 3T5b code ($n = 5, k = 1, \nu = 27$). Black edges represent bipartite assignments, purple dotted lines are the virtual edges for a current location to assign to all available code words. **Left**: start of episode, **center**: middle of episode, **right**: end of episode after all locations have been assigned.

### 3.5 REWARD DESIGN

We designed reward functions that encourage the agent to optimize assignments based on gate count reduction. For value-based methods like DQN, the reward at step $i$ is $R_i = \alpha \cdot (\gamma + GC_{i-1} - $

$GC_i)^\beta$ where $GC_i$ is the gate count at step $i$, $\alpha$ adjusts the magnitude, $\beta$ controls sensitivity, and $\gamma$ sets a baseline for reduction.

For policy-gradient methods like PPO, we compute post-episode rewards based on final synthesis results: $R_i = 1 - \sum_m \frac{1}{2^{d_m}}$ where $m$ is a minterm including assignment $i$, and $d_m$ is the degree of the minterm, rewarding more shared terms in the final synthesis. Both structures incentivize codeword assignments that minimize gate count through shared logic terms.

## 4 RESULTS

The experiments were conducted on GPUs using a NVIDIA-DGX2 system with the PyTorch container image (nvcr.io/nvidia/pytorch:24.01-py3). We evaluate the RL algorithms discussed in Section 3 and provide ablation studies compared against baseline heuristics including the similar L1 similarity feature. To demonstrate the generalizability of our framework, we show results for a number of different constrained codes: the MTA 7b8b code mentioned in Section 1, 5s8b which is larger trinary to binary code, and an 8b9b and an 8b9b code which mitigates crosstalk in high-speed links Sudhakaran & Newcomb (2016). While most ablation studies focus on the MTA 7-8 bit code, we show comparative results for all of the codes.

### 4.1 HYPERPARAMETER AND ALGORITHMIC COMPARISONS

We start by evaluting the impact of different hyperparameters and learning approaches on the performance of our Double-DQN models on the MTA code. The results are presented in Fig. 8, which highlights the performance across three different sweeps: learning and target network update rate.

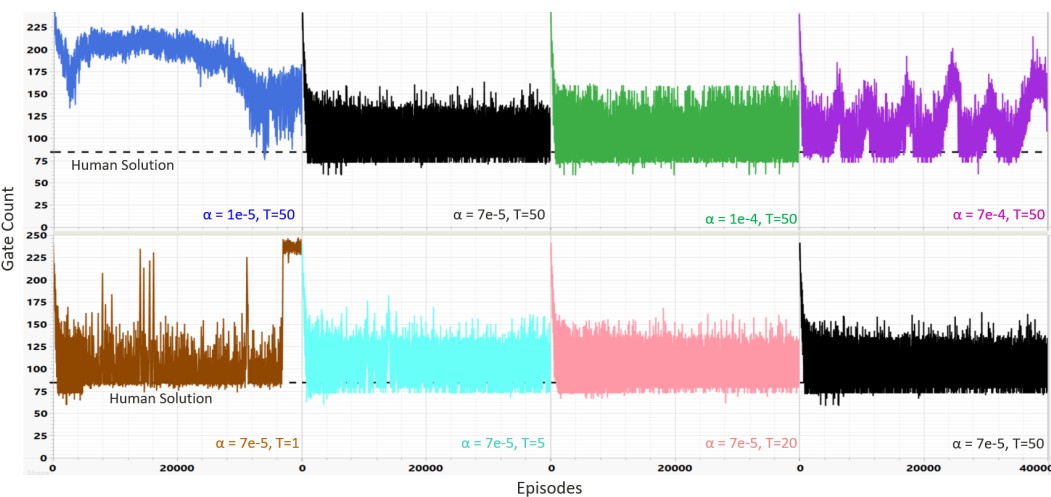

Figure 8: Double DQN Results Across Selected Hyperparamter Sweeps: Double-DQN Target Network Update Rate. Top Row Learning Rate Sweep with ($T = 50$) and Bottom Row Target Update Rate Sweep with ($\alpha = 7e{-}5$)

The first row illustrates the results of a sweep over the learning rate $\alpha$, where values of $\alpha = 1e^{-5}$, $7e^{-5}$, $1e^{-4}$, and $7e^{-4}$ while keeping the target network update rate constant ($T = 50$). Higher learning rates (green, purple) converge faster, but the $7e^{-4}$ case exhibits wild fluctuations. We also noticed that very low learning rates converged slower (blue). In the second row, we explore the effect of varying the target network update rate $T = 1, 5, 20, 50$ while keeping $\alpha$ constant ($\alpha = 7e^{-4}$). As shown, faster target rates (brown) yield higher fluctuations. For the remainder of the Double-DQN cases, we chose the control case (black) parameters ($\alpha = 7e^{-5}, T = 50$).

Next, we compare the rolling minimum gate count across a several RL algorithms. In Fig. 9(a), we present the distributional RL algorithm (C51) with varying number of quantiles: $c = 7, 9, 11, 25$. While the final minimum gate counts converge similarly, lower quantile cases exhibited a lot more

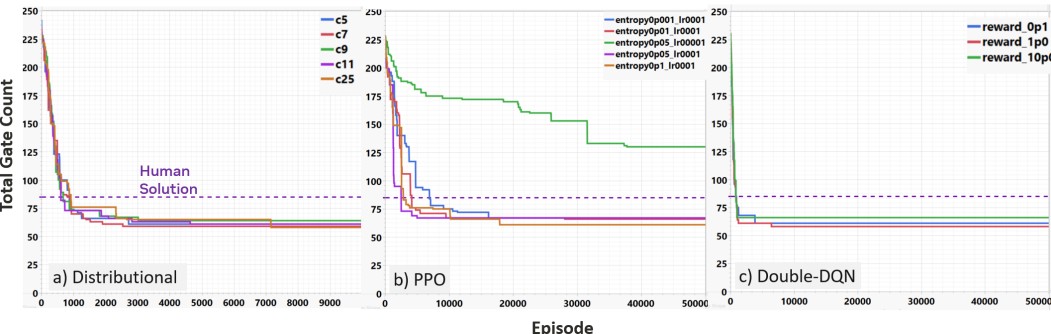

Figure 9: Rolling Minimum Gate Count Across RL Algorithms and Parameter Sweeps (a)Distributional RL (b) PPO (c) Double-DQN

fluctuation albeit not visible in the rolling minimum plots. In subplot (b) we compare the PPO algorithm across various entropy and learning rate parameters. Over more episodes, the PPO algorithms eventually attained similar minimum gate counts outside of the case with the $1e^{-5}$ learning rate. In subplot (c) we show the double-DQN algorithm with different reward linear scaling factors ($\alpha$). Double-DQN reached the lowest minimum gate count (57), the fastest time to attain it, and lowest variation. Moving ahead, we focus on using Double-DQN with $\alpha = 1.0$.

## 4.2 NETWORK ARCHITECTURE ABLATION STUDIES

We experimented with the GNN and L1 similarity components to evaluate their impact on both performance and resource usage, as GNNs typically add extra computational and memory demands Hamilton et al. (2017). As shown in Fig. 10, the model without the GNN (green) still managed to learn and approach a somewhat similar gate count, but it took a lot longer to converge (15K vs. 2K episodes with the GNN). Additionally, the memory footprint of the GNN model was much higher, requiring 1177MB compared to 136MB for the non-GNN version. The L1 similarity heuristic also proves to be quite critical. Without it (red), the model struggles and plateaus at a much higher minimum gate count showing that domain-specific knowledge dramatically improves performance. Thus the integration of both the GNN and L1 similarity achieve faster convergence and better solutions in this setting, but the ablation study without the GNN shows promise and can benefit from further study to keep the memory footprint down when scaling to larger codes.

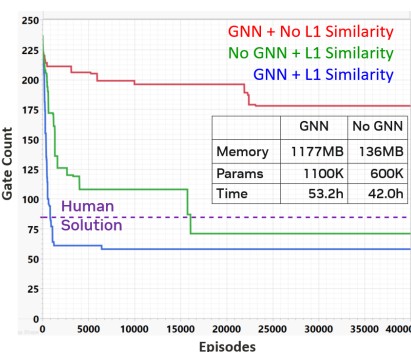

Figure 10: Ablation Study: Removing GNN, L1 Similarity

## 4.3 RL PERFORMANCE AGAINST HEURISTICS ACROSS CODES

Now we compare the performance of our RL framework across the three codes against heuristics guided with the same L1 similarity algorithm. Figure 11 shows the rolling minimum gate count of our best performing RL configuration against the simulated-annealing approach guided by the L1 similarity algorithm. Subplot (a) shows the MTA code followed by the 8b9b code in (b) and 5s8b code in (c). In all three cases, there is an appreciable improvement in RL over guided heuristics.

## 4.4 LEARNING FROM THE RL AGENT'S BEST TRAJECTORIES

To gain insight into the RL agent selection algorithms, we looked at the trajectories of a few cases in one of our double DQN training cases. Figure 12 illustrates the solution trajectories, highlighting

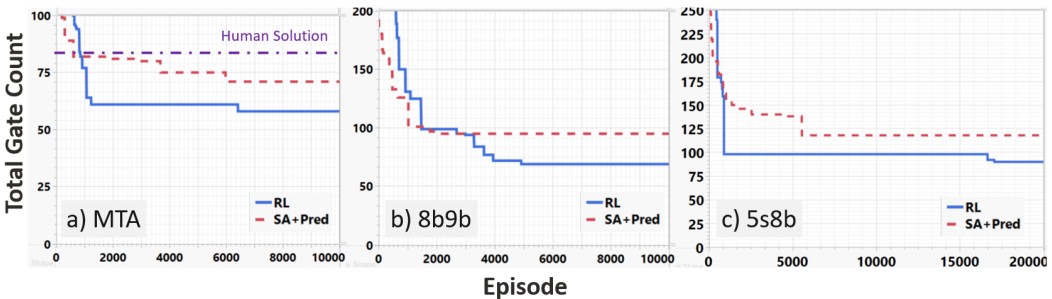

Figure 11: Rolling Minimum Gate Count Across Codes (a) MTA, (b) 8b9b, (c) 5s8b

the gate count increase per episode step for three solutions: a poor policy at the begining of training, a mediocre solution in the middle of training, and the best solution achieved during training. For each point, the gate count for both encoder and decoder is plotted across assignment steps, allowing us to visualize the trajectory of complexity.

The most notable observation is the nearly flat portions of the trajectories between steps 32 and 95 in the best solutions (blue, orange), across each code. This suggests that the learned policy effectively minimizes additional complexity during these steps, largely through extensive term sharing. Notably, the agent appears to replicate rules for location-to-codeword mappings during this segment, taking advantage of the changes in the most-significant bits (MSBs) in the locations. Specifically, the agent efficiently assigns similar codes where the MSBs differ (32-63 = "01" and 64-95 = "10"), resulting in minimal complexity increase. However, when the MSBs change to "11" for locations 96–127, corresponding to a restriction in the MTA code (where the first symbol cannot be "11"), the agent is forced to introduce additional complexity to adhere to this rule. In contrast, the mid-level and initial solutions show steadily increasing gate counts, reflecting suboptimal policies with little to no term sharing.

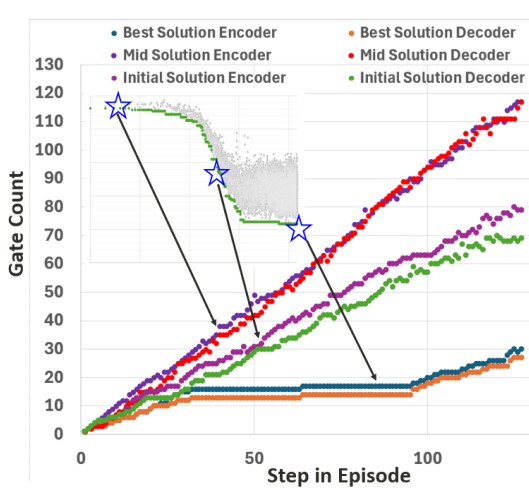

Figure 12: Training Data and Trajectories for Initial, Mid, and Best Solutions

We expanded this analysis to all three codes. Figure 13 demonstrates the cumulative gate count progression for these codes, paired with visualizations of the location and codeword assignments for the best solution found by the agent. To better illustrate the learned term-sharing patterns, we have highlighted key block structures in green within the rightmost black-and-white visualizations. These blocks reveal where the agent replicates specific assignment rules to optimize term sharing, particularly in sections where the most-significant bits differ. This visualized replication indicates the agent's ability to generalize its approach across different constrained codeword mappings. Across all codes, the agent consistently discovers policies that minimize gate count through careful and structured assignments.

## 5 LIMITATIONS

While our RL framework demonstrates strong performance on encoding schemes with varying input space dimensions, scaling it to even larger codes presents challenges. Specifically, we encountered memory and time constraints due to the larger search space of these codes. Ongoing work is focused

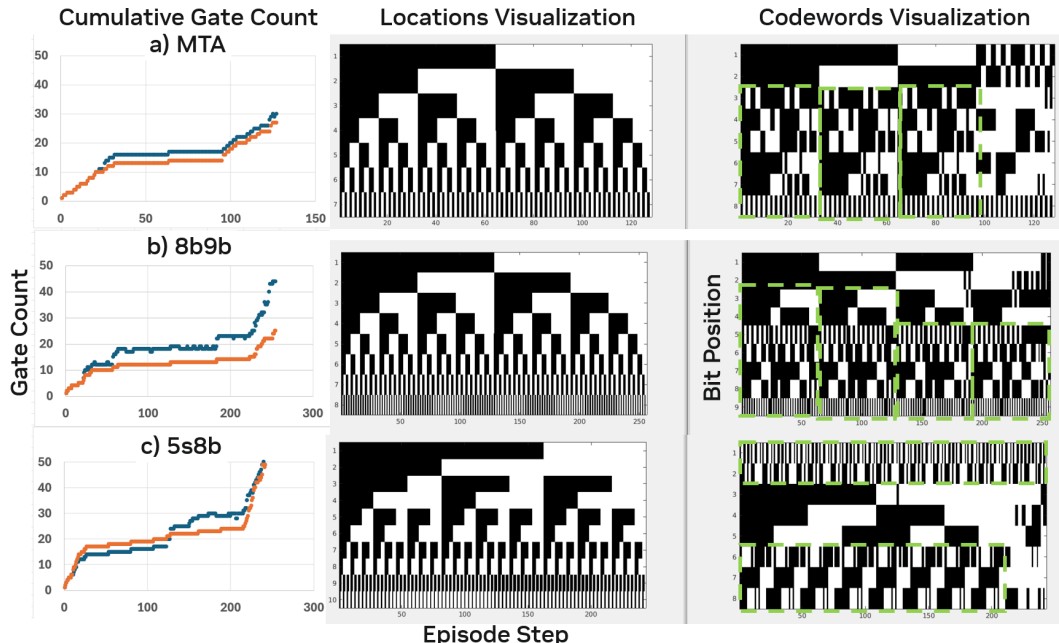

Figure 13: Cumulative Gate Count & Binary Visualization of Code Assignments for Best Episode

on exploring memory-efficient techniques, such as hierarchical RL, and leveraging policy distillation from good trajectories to develop policies for larger codes.

Another limitation is that our current approach uses gate count as the primary proxy for logic complexity, which stems from the use of the open-source synthesis tool Espresso Brayton et al. (1984). Larger codes and more complex designs may require alternative synthesis methods to optimize circuit efficiency beyond Sum-of-Products (SoP) representations. We did explore using ABC, a more powerful synthesis tool than Espresso; however, ABC does not support features such as "don't cares" in the lookup table, which limits the agents' ability to find optimal solutions. Further details on this exploration are provided in the appendix.

Finally, while out current reward function has been effective, we did not fully explore alternative reward schemes. Ongoing work investigates different reward configurations, which may reveal more efficient pathways for optimization and better convergence in larger-scale problems.

## 6 CONCLUSION

In this work, we introduced a **reinforcement learning (RL) framework** to automate the process of constrained code design, addressing a critical gap left by synthesis tools that cannot optimize input mappings. Our approach was applied to challenging encoding schemes, including **MTA (7b8b), 8b9b, and 5s8b (ternary weight compression) codes**. By leveraging an **L1 similarity-based heuristic**, our RL framework efficiently explores the solution space using feedback on key metrics such as gate count, outperforming random search and hand-crafted designs.

Our results demonstrate that **RL can handle complex encoding schemes**, providing a practical solution for automating hardware optimization tasks. Furthermore, beyond constrained codes for high-speed links, this approach shows promise for **efficient quantized AI model weight storage**, an area of growing interest. While this work demonstrates the effectiveness of our RL framework on a variety of constrained code designs, ongoing refinements are aimed at scaling the framework to handle even larger codes and exploring additional heuristics to further improve computational efficiency and solution quality.

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

## A  APPENDIX

### A.1  SYNTHESIS TOOL SELECTION

We experimented with both ABC and Espresso for back-end synthesis tools, but elected to use the latter due to its computational speed and efficiency. The SoP implementation results in a circuit with only two levels of logic: one level for the AND gates (products) and one for the OR gate (sum of products). This structure inherently limits the delay or levels of logic, making the gate count and area the primary factors affecting the QoR. Given this, our optimization objective simplifies to minimizing the gate count, as delay is no longer a significant concern in a two-level SOP implementation: $min_{A \subseteq X \times Y} \text{GateCount}(A)$.

### A.2  MLP DESIGN

The structure of the MLP described in the Network Architecture section consists of three fully connected layers:

- **Layer 1:** A fully connected layer with an input dimension corresponding to the binary input size and output dimension of 256. This layer is followed by a LayerNorm operation and a ReLU activation function.
- **Layer 2:** A fully connected layer doubling the hidden dimension to 512, followed by another LayerNorm and ELU activation.
- **Layer 3:** A final fully connected layer reducing the hidden size back to 256, with LayerNorm and ELU applied. A dropout of 30% is applied between each layer to prevent overfitting.

## A.3 GAT CONFIGURATION

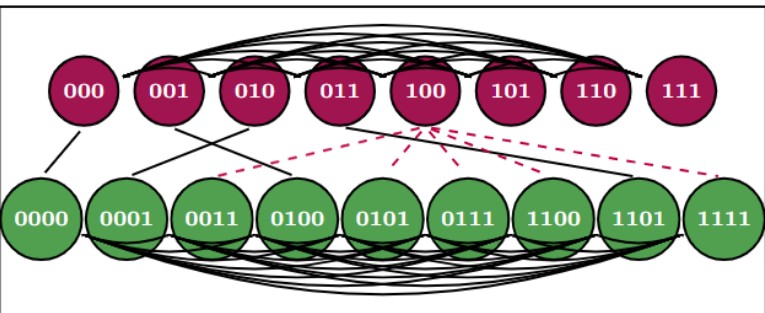

Figure 14: Graph Structure for GAT Input

Figure 14 shows the graph representation for the GAT for a 3-bit 2-Ternary (3b2T) code example. First, we define intra-set edges among all pairs of nodes in $X$ and $Y$ respectively. We experimented with and without weights on these edges representing the reciprocal L1 similarity (hamming distance). Earlier assignments in the episode between location and code nodes are maintained as bipartite edges. Virtual edges, representing all possible assignments of the current location node to available code nodes, are shown as dashed lines. We include these edges for the network to help influence the Q-values.

## A.4 ALGORITHMIC TECHNIQUES

The LUT formulation is a rather straightforward approach to implement encoder and decoder functions, but other techniques have been published over the last few decades. For instance, Adler et al. (1983) pioneered the development of sliding block codes in the 1980s, utilizing a finite state machine (FSM)-based approach to enforce constrained codes like run-length limited (RLL) codes briefly discussed in the introduction. Their method, while effective for defining and enforcing constraints, involves design decisions such as state splitting and transition assignments, which can impact the hardware efficiency of the implementation. Our work, while distinct, is complementary to this approach. Namely, our methodology offers potential optimizations that could be applied to the state splitting and transition assignments in Adler's FSM-based designs.

