# OpenReview forum: "Reinforcement Learning and Heuristics for Hardware-Efficient Constrained Code Design"
_ICLR.cc/2025/Conference — Submitted to ICLR 2025_

### Official Review · Reviewer_zmR7 · 2024-10-24

**Soundness:** 3
**Presentation:** 2
**Contribution:** 2
**Rating:** 5
**Confidence:** 4

**Summary:**

This paper proposed a RL-based approach to find the optimal one-to-one mapping between a location space ($\\{0, 1\\}^n$) and a codeword space ($\\{0, 1\\}^{n+k}$), with respect to the required number of gates to implement the mapping. One episode of the RL process contains $2^n$ steps, and the $i$-th step assigns an available codeword to a location. The reward in each step is the difference of gate count between step $i-1$ and step $i$. A crafted feature based on L1 similarities plays a critical role to the performance, and the inclusion of GNN also boosted the performance with a cost of additional time/memory/parameters.

**Strengths:**

1. This paper provides an approach that directly deliver an optimal mapping through RL, which is usually more challenging than incremental improvement of existing heuristics or exact methods (especially w.r.t performance).
2. This paper addressed a real-world problem in communcation and digital design.
3. The L1 heuristics is well crafted (especially for area optimization).
4. Some ablation studies are provided.

**Weaknesses:**

1. The evaluation seems not very sufficient. No other baseline methods in other papers are compared. The only comparison happens at Section 4.3, with simulated annealing guided by the L1 heuristics proposed in this paper (in line 258-260 the paper also mentioned greedy search but I cannot find it), and the details / code are not provided. See the "Questions" section for details.
2. The trained model is code specific, so different models need to be trained for different codes. For example, a model trained on MTA code cannot be transferred to other codes like 8b9b/5s8b.
3. The model heavily relies on a hand-crafted L1 heuristics to achieve decent performance.
4. Equation (1) of the paper mentioned latency and power consumption as the objective, but this is not reflected in the rest of the paper.
5. The writing can be largely improved. E.g., the definition of state, action and reward is not clear enough. See the "Questions" section for details.

**Questions:**

Questions:
1. Is there any other paper addressing similar problems as in this paper (finding valid codeword assignments that can be implemented by circuits with optimal PPA)?
2. What does "Pred" mean in Figure 11?
3. What is the x axis for simulated annealing in Figure 11?

Suggestions:
1. Provide a formal definition of state, action and reward (better with mathmatical notations)
2. If the problem addressed in this paper has been tackled in other papers, consider citing and comparing with them. If not, consider highlighting this fact in the introduction section ("we are the first to ..."), maybe with some intuition why this problem has not been tackled in previous literatures.
3. Consider providing the detail/code of the baseline methods (simulated annealing, ...) in the appendix/supplemental material.
4. Consider using wall time as x axes for Figure 11. RL and simulated annealing are basically very different algorithms.
5. Consider adding more baselines if possible.
6. Consider simplifying/removing some content that barely contribute to this paper. E.g., I don't think line 48-52 is so relevant to this paper. Also, Section 3.1 and 4.1 may be simplified or moved to the appendix, leaving space for more thorough, detailed discussion of the proposed method (for example, I would be interested in the intuition behind step 3 of algorithm 1, and the role of "virtual edges" in Section 3.4).

Moreover, it will be more appealing if the model can perform decently even without an explicitly given heuristics (i.e., learn a decent heuristics by the model itself).

---

### Official Review · Reviewer_PUrJ · 2024-11-02

**Soundness:** 2
**Presentation:** 2
**Contribution:** 2
**Rating:** 3
**Confidence:** 5

**Summary:**

This paper presents a reinforcement learning (RL) framework to automate the process of constrained code design. The proposed RL strategy has been applied to encoding schemes, including MTA (7b8b), 8b9b, and 5s8b (ternary weight compression) codes. By leveraging an L1 similarity-based heuristic, the authors state that their proposed RL framework efficiently explores the solution space using feedback on key metrics such as gate count, outperforming random search and hand-crafted designs. The paper does not make any theoretical contributions to RL but showcases how RL can be used for constrained code design.

**Strengths:**

+ Solving constrained code design problem can have some applications in communication and data storage

**Weaknesses:**

- It is unclear how the encoding scheme and other codeword and compiler approaches influence the chosen L1 heuristic RL
- It is unclear how this RL framework need to be modified to handle multi-objective optimization including timing, power, and area simultaneously
- Minimal if any analysis on scalability and comparison with possible alternative approaches.

**Questions:**

0. How does the encoding scheme and other codeword and compiler approaches influence the chosen L1 heuristic?
1. The paper heavily relies on the L1 similarity heuristic for optimization, but how does this approach scale with significantly larger code spaces? Is there a more efficient way to incorporate domain knowledge without the computational overhead of calculating pairwise similarities across all potential mappings?
2. What is the reasoning behind adopting this L1 similarity heuristic? Why not any other similarity metric?
3. While the paper focuses on gate count as the primary optimization metric, how would the RL framework need to be modified to handle multi-objective optimization including timing, power, and area simultaneously? What are the tradeoffs between these different objectives in constrained code design?
4. The proposed GNN architecture improves convergence speed but increases memory usage significantly. Could alternative graph neural network architectures like GraphSAGE or GAT variants provide better memory efficiency while maintaining the convergence benefits? How would this impact scalability to larger codes?
5. The paper demonstrates success with Double-DQN but only shows limited results with PPO. How would more advanced policy-based methods like SAC or TRPO perform on this task? Could hybrid approaches combining value and policy optimization provide better results?
6. The current framework uses synthesis tools as black-box optimizers. How could the RL framework be enhanced to incorporate knowledge of the synthesis process itself? Could this lead to more targeted optimization strategies that better guide the code assignment process?

---

### Official Review · Reviewer_pBM2 · 2024-11-02

**Soundness:** 3
**Presentation:** 2
**Contribution:** 1
**Rating:** 5
**Confidence:** 3

**Summary:**

The authors propose a reinforcement learning-based framework for finding mapping for constrained codes. The RL algorithm utilizes the proposed L1 similarity heuristics when searching for the mappings. A reward function that includes the gate count is provided during training to induce the network to produce efficient mappings. Experimental results show that the performance of the proposed RL algorithm is superior to that of human-crafted solutions in terms of circuit complexity obtained using the Espresso tool.

**Strengths:**

The paper proposes to utilize RL for constrained codes, and the idea seems novel and exciting. The proposed ideas are significant to the community. For the most part, the paper is well-written and easy to follow.

**Weaknesses:**

1. The first major weakness for me is the justification of using the RL agent for this task. The authors provide a comparison of the proposed  RL model with the simulated annealing algorithm (SA) that utilizes the L1 similarity score. However, the SA algorithm does not use circuit complexity in its objective, making the comparison unfair for the simulated annealing algorithm. The authors need to include this in the updated paper. If the SA algorithm is performing well, the paper is still novel, just not suited for this venue.
2. No comparison with existing automated mapping algorithms has been made. The only comparison has been with human solutions.
3. Claims such as "we propose applying constrained codes to further compress ultra-quantized AL models" should be removed from the paper as no experimental results are provided to validate this claim.
4. Most of the figures in the paper have the following problem:
(a) The captions are too short, and consequently, it is difficult to find what the figures are representing. For example, "Figure 1: RLL Constrained Code Illustration" should be changed to provide a bit more information.
(b) The figures themselves look off; the text can be too small, or certain texts are cut off.  Figures 9 and 11 have text that has been cut off. It is difficult to read Figure 7, as the text is too small. These issues are widespread and not just seen in the examples cited above.

**Questions:**

None.

---

### Official Review · Reviewer_tEqD · 2024-11-03

**Soundness:** 3
**Presentation:** 2
**Contribution:** 2
**Rating:** 5
**Confidence:** 4

**Summary:**

The paper first starts by delving into the fact that Constrained Codes are important in enhancing reliability and optimizing bit efficiency in high-speed communication systems. However, a major challenge in designing these codes is minimizing the hardware complexity of the translation logic used for encoding and decoding. The paper then explores an reinforcement learning based approach that outperforms manual solutions.

The work sets the objective function to optimize area, latency, and power which the proposed framework utilizes the feedback from logic synthesis tools. The framework employed an Online (live-feedback to adapt to various problem) / Model-free (due to the intractable non-linear solution space) / both Value-based and Policy-based RL. Important recipe proposed in the work is "domain-specific heuristic" of L1 similarity metric.

It seems that the paper illustrates a concrete use-case of RL with ample experiments. However, the paper needs work in presentation and seems like there are questions that need to be answered. Overall, I am in between 3: reject, not good enough and 5: marginally below the acceptance threshold, but leaning more to marginally below acceptance threshold.

**Strengths:**

It seems that the paper illustrates a concrete use-case of RL with ample experiments.

**Weaknesses:**

The paper can work on improving the presentation. For example the references to the figures

The paper seems falls short in providing insights that can scale outside the given problem. Potentially this may be improved by restructuring the paper to spend a little more effort describing the problem to the people who are new to Constrained Codes instead of relatively less important 2.1-2.3.

Also, while the paper provides "a" design point given the problem it seems that the paper does not seems to provide enough insights as to why L1 similarity heuristic is really needed. It seems that there are more questions casted by reading 3.2 than answered.

**Questions:**

* Can you provide more insight to why L1 similarity and heuristic are needed? Besides the argument that "it converges better", why can't we rely on end-to-end neural approach?
* How does the work compare to program synthesis works such as [1] and [2]. I believe there has been large advances in the area to build the smallest programs that can create code for functions that map X -> Y. The example given in Figure 5 seems to be very similar to the problems discussed there.

[1] Massalin, Henry. "Superoptimizer: a look at the smallest program." ACM SIGARCH Computer Architecture News 15.5 (1987): 122-126.
[2] Lee, Woosuk, et al. "Accelerating search-based program synthesis using learned probabilistic models." ACM SIGPLAN Notices 53.4 (2018): 436-449.

* Can you please add references to Figure 2, 5, 7-11, in the text? This is very important to improve readability

---

### Official Review · Reviewer_mnsE · 2024-11-04

**Soundness:** 3
**Presentation:** 3
**Contribution:** 3
**Rating:** 6
**Confidence:** 4

**Summary:**

This paper presents a novel framework for designing hardware-efficient constrained codes, which are crucial for optimizing bit efficiency and reliability in high-speed communication systems. The authors apply reinforcement learning enhanced by an L1 similarity-based heuristic to automate the design of encoding and decoding logic, typically a time-intensive, manual process. By framing the codeword assignment as a bipartite graph matching problem, the proposed RL approach efficiently navigates the vast design space, outperforming traditional human-designed and heuristic solutions in reducing hardware complexity, specifically in metrics like gate count, area, and power.

**Strengths:**

1.	The paper introduces a unique application of RL and a similarity heuristic to a problem in hardware design, innovatively applying RL to optimize constrained code design.
2.	The framework is extensively tested across multiple encoding schemes, demonstrating generalizability and outperforming both random search and existing heuristic approaches.
3.	The ablation studies and comparison tables effectively highlighting results.

**Weaknesses:**

1.	Although the RL framework performs well on the tested encoding schemes, the paper acknowledges limitations in scaling the approach to larger codes due to memory and computational constraints.
2.	The reward function is primarily focused on minimizing gate count, which can limit the model’s exploration of multi-objective optimizations, such as balancing gate count with delay.
3.	The reliance on the L1 similarity heuristic suggests that the RL model may not generalize well to problems where this heuristic is less applicable.
4.	The layout and resolution of the figures could be improved to enhance clarity and presentation quality.

**Questions:**

1.	How would the proposed RL framework perform on larger codeword assignment problems, particularly in terms of memory and time requirements?
2.	The paper’s reliance on an L1 similarity heuristic is effective but may limit generalization. Have you considered other heuristics (e.g., Hamming distance, cosine similarity) or data-driven heuristics for prioritizing assignments?
3.	While gate count is the primary metric in the reward function, other hardware metrics (like delay) could also impact final performance. Could the proposed framework enhance the balance across multiple hardware constraints?

---

### Meta-Review · Area_Chair_n8BL · 2024-12-13

**Metareview:**

This paper introduces an RL framework improved with an L1 similarity-based heuristic for designing hardware-efficient constrained codes, focusing on metrics like gate count, area, and power. The authors frame the task as a bipartite graph matching problem and demonstrate improved performance over human-designed and heuristic solutions. While the problem tackled is relevant, and the results are promising for the evaluated encoding schemes, the paper falls short in the following key areas. The RL approach heavily relies on the L1 similarity heuristic, raising concerns about generalizability to broader problem spaces. The paper also lacks comparisons with existing state-of-the-art automated mapping methods, and the scalability of the proposed method is not convincingly addressed (benchmarking). Additionally, the presentation could be improved, with clearer explanations of the methodology and better integration of figures and results into the narrative. The lack of a rebuttal from the authors to clarify or address these points further impacts the assessment. Considering these factors, I recommend rejecting this submission.

**Additional Comments On Reviewer Discussion:**

The reviewers raised concerns about the heavy reliance on the L1 similarity heuristic, the lack of generalizability to larger or different constrained codes, and limited multi-objective optimization (e.g., balancing area, power, and delay). They also highlighted the absence of comparisons with existing automated mapping techniques and issues with the paper's clarity and presentation. During the rebuttal phase, the authors did not provide a response to address these concerns or clarify ambiguities. Without further discussion or additional evidence from the authors, the reviewers' concerns remained unaddressed, leading to the conclusion to recommend rejection for this paper.

---

### Decision · Program_Chairs · 2025-01-22

Reject